# Novel Strategies for Solubility and Bioavailability Enhancement of Bufadienolides

**DOI:** 10.3390/molecules27010051

**Published:** 2021-12-22

**Authors:** Huili Shao, Bingqian Li, Huan Li, Lei Gao, Chao Zhang, Huagang Sheng, Liqiao Zhu

**Affiliations:** College of Pharmacy, Shandong University of Traditional Chinese Medicine, 4655 Daxue Road, Jinan 250355, China; shl20200926@163.com (H.S.); Lbq993722@163.com (B.L.); lihuan1459672316@163.com (H.L.); gaolei11252021@163.com (L.G.); tougaotcm@163.com (C.Z.)

**Keywords:** bufadienolides, solubility, bioavailability, structural modification, nanoformulation, pharmacological activity

## Abstract

Toad venom contains a large number of bufadienolides, which have a variety of pharmacological activities, including antitumor, cardiovascular, anti-inflammatory, analgesic and immunomodulatory effects. The strong antitumor effect of bufadienolides has attracted considerable attention in recent years, but the clinical application of bufadienolides is limited due to their low solubility and poor bioavailability. In order to overcome these shortcomings, many strategies have been explored, such as structural modification, solid dispersion, cyclodextrin inclusion, microemulsion and nanodrug delivery systems, etc. In this review, we have tried to summarize the pharmacological activities and structure–activity relationship of bufadienolides. Furthermore, the strategies for solubility and bioavailability enhancement of bufadienolides also are discussed. This review can provide a basis for further study on bufadienolides.

## 1. Introduction

Toad venom (Bufonis Venenum, known as ‘Chansu’ in Chinese), the dry secretion of *Bufo bufo gargarizans* Cantor and *Bufo melanostictus* Schneider, is recorded in the 2020 edition of the Chinese Pharmacopoeia and has efficacy in providing detoxification, pain relief, resuscitation and refreshment. It can be used in the treatment of carbuncles, gangrene, sore throats, sunstroke, abdominal pain, vomiting and diarrhea [1]. The chemical composition of toad venom includes bufadienolides, indole alkaloids, sterols, polysaccharides, amino acids and organic acids, among which bufadienolides and indole alkaloids are considered to be the main active ingredients [2]. *Bufo bufo gargarizans* Cantor and *Bufo melanostictus* Schneider squeeze out a white slurry from their posterior auricular glands and skin glands, which is processed and dried to obtain toad venom. After the toad venom is extracted, the toad can recover in three to four weeks (Figure 1). Bufadienolides are the main chemical components of toad venom, whose structures are steroids connected with a α-pyrone ring at position C-17 [3]. Studies have shown that toad venom contains 96 kinds of bufadienolides monomers [4], among which gamabufotalin (CS-6), arenobufagin (ABG), 1β-hydroxyl-arenobufagin (1β-OH-ABF), telocinobufagin (TBG), bufalin (BF), hellebrigenin (HBG), ψ-bufarenogin and 19-hydroxybufalin (19-HB), bufotalin (BL), cinobufotalin (CBL), cinobufagin (CBG), resibufogenin (RBG) and marinobufagin (MBG) are considered the major effective anticancer agents (Figure 2), and these ingredients have high activity and high toxicity profiles [5,6,7]. Additionally, bufadienolides have many other pharmacological activities in vivo and in vitro, such as cardiovascular, analgesic and anti-inflammatory effects, and so on. The content of CS-6, TBG, BL, CBL, BF, CBG and RBG accounts for more than 85% of the total bufadienolides. BF, CBG and RBG account for a large proportion of bufadienolides compounds, and the total content of the three compounds is 5–10% [8]. Cao et al. [9] found that there was a correlation between CBG, BF and TBG and the total content of bufadienolides. Compared with the existing matrix control indices of CBG and RBG, these three components better reflect the overall content of bufadienolides. This discovery is helpful in the revision of the quality standard of toad venom.

Bufadienolides exhibit multiple pharmacological activities, but their low solubility and bioavailability limit clinical applications. Liu et al. [10] used high-performance liquid chromatography and the shake flask method to determine the apparent solubility and apparent oil–water partition coefficients of RBG, CBG and BF. The results showed that the apparent solubility of RBG, CBG and BF was 76.29 μg/mL, 51.85 μg/mL and 32.76 μg/mL at 37 °C (pH = 7.0), respectively. The oil–water partition coefficients of all three were large, typical hydrophobic products, and the hydrophobicity size order was RBG > BF > CBG. These results provided a basis for adopting strategies to improve the solubility and bioavailability of bufadienolides. Because of its structural similarity, the water solubility of bufadienolides was poor and the hydrophobicity was extremely strong.

In order to solve the problems of low solubility and bioavailability of bufadienolides, scholars have done a lot of research. Among the various strategies, structural modification and nanodelivery systems have shown remarkable achievements in improving the solubility and bioavailability of bufadienolides. In this review, we aim to summarize the pharmacological activities and structure–activity relationship of bufadienolides. Furthermore, the strategies of improving the solubility and bioavailability of bufadienolides by structural alterations and the establishment of drug delivery systems in recent years are summarized.

## 2. Bufadienolides’ Structure–Activity Relationship

Bufadienolides are an important class of polyhydroxy C-24 steroids and their glycosides. They are characterized by the presence of a six-membered lactone (α-pyrone) ring at the C-17β position and exhibit antitumor properties [11]. However, the substituents at C15 and C16 are detrimental to the growth inhibition of cancer cells [12]. Azalim et al. observed that the cyclization of C14–15 of bufadienolides caused a dramatic decrease in the affinity for binding to Na^+^/K^+^-ATPase in the E2P conformation and accelerated the rate of inhibition of K^+^-pNPPase, and the carbohydrate free moiety in C3 increased the rate of inhibition [13]. Cytotoxicity date showed that the C3 moiety had an important effect on the cytotoxicity activity of bufadienolides [14]. From the perspective of structure–activity relationship studies, the 16-acetoxyl group and 14β-hydroxyl group are crucial for bufadienolides activity, thus the disappearance of the 14β, 15β-epoxy ring increases their cytotoxicity [15]. In addition, among the relative configurations of bufadienolides rings A, B, C and D, compounds in the C-ring and D-ring cis-configuration are the most potent cancer cell growth inhibitors [16]. The structure–activity relationship of bufadienolides is illustrated in Figure 3.

## 3. Pharmacological Activities of Bufadienolides

Recently, bufadienolides have been explored for multiple pharmacological activities, including antitumor, cardiovascular effects, anti-inflammatory, analgesic and immunomodulatory. The pharmacological activities of bufadienolides are presented in Table 1.

### 3.1. Antitumor

Bufadienolides have antitumor activity and are a potential candidate for the treatment of cancer. Studies have shown that BF has antitumor effects of inhibiting cell proliferation and inducing tumor cell apoptosis [17,18,19]. BF could promote proteasome activation and ATP1A1 protein degradation to inhibit ATP1A1 expression and significantly inhibit tumor growth in glioblastoma [20]. The inhibition of neuroblastoma cell proliferation and migration by BF was dose-dependent and time-dependent [21]. ABG and HBG could downregulated the expression levels of Cdc25C and Cyclin B1, reduce the cell survival rate. Both of them exhibited distinct cytotoxicity against cancerous glial cells with high potency, selectivity and tolerability [22]. Zhang et al. [23] studied the antitumor effects of HBG and ABG on estrogen receptor (ER)-positive breast cancer cell line MCF-7 and triple-negative breast cancer cell line MDA-MB-231. HBG showed stronger cytotoxicity than ABG in both cancer cells. The results indicated that HBG and ABG were potentially useful in developing treatment strategies to treat different types of breast cancer, especially ER-positive breast cancer. Deng et al. found that 1β-OH-ABG from bufadienolides could inhibit the increased proliferation of hepatoma cell lines Hep3B, HepG2, HuH7 and SK-HEP-1, but had less cytotoxic effect on normal hepatocyte LO2. The mechanism was that 1β-OH-ABG could obviously reduce the expression levels of p-AKT/AKT and p-mTOR (Ser2248 and Ser2481)/mTOR in a time-dependent manner [24].

Studies have shown that MBG, TBG, BF and CBG have a significant inhibitory effect on human tumor cells. The mechanism is similar to that of cardiac lactones in inhibiting Na^+^/K^+^-ATPase. These compounds have high selectivity for tumor cells, and can reduce the level of anti-apoptotic proteins, inhibit DNA synthesis, arrest the cell cycle in G_2_/M or S phase and depolarize mitochondria [25]. Yang et al. [26] demonstrated that CBG reduced the proliferation and colony forming ability of human liver cancer cells in vitro, and induced mitotic arrest of human liver cancer cells. CBG could reduce the expression of EGFR and CDK2 activity in human liver cancer cells, thereby inhibiting the growth of tumor cells, and enhancing the inhibitory effect of CBG on the proliferation potential of human liver cancer cells.

According to Ding et al. [27,28], ψ-bufarenogin inhibited the proliferation of liver cancer cells by blocking cell cycle transition, and promoted apoptosis by downregulating the expression of Mcl-1. Moreover, ψ-bufarenogin reduced the number of liver cancer stem cells by inhibiting Sox2, and had a synergistic effect with conventional chemotherapy drugs. 19-HB could inhibit the proliferation, migration and invasion of non-small cell lung cancer (NSCLC) cells and promote the apoptosis of NSCLC cells through the Wnt/β-catenin pathway. In addition, 19-HB inhibited the growth of xenograft tumors in nude mice and had little toxicity to the liver and kidney. Therefore, 19-HB may be a potential antitumor drug for the treatment of NSCLC [29].

Bufadienolides not only have strong antitumor effects by themselves, but also exert antitumor activity in combination with other antitumor agents. The combination of multiple agents can significantly improve the efficacy of drugs, reducing drug resistance and toxic side effects. HER2 is a proto-oncogene frequently amplified in human breast cancer, and its overexpression is related to tamoxifen resistance and reduced recurrence-free survival. Studies showed that ABG and BF could significantly inhibit the proliferation and survival of HER2 overexpressing breast cancer cells, and downregulate the expression of SRC-1, SRC-3, nuclear transcription factor E2F1, phosphorylated AKT and ERK. The combination of low-dose bufadienolides and tamoxifen could significantly enhance the inhibitory effect of tamoxifen on HER2 overexpression breast cancer cells, which suggested that ABG and BF might be potential adjuvant therapy drugs for the treatment of HER2 overexpression breast cancer [30]. Paclitaxel (PTX) is an effective drug for the treatment of cancer [31]. A binary mixture of PTX+CBG and BF+CBG was shown to have a synergistic effect on cancer cells. A ternary mixture of PTX+BF+CBG had a strong synergistic effect on cancer cells. The combination of BF and CBG synergistically enhanced the efficacy of PTX on HepG2 cells, and the combination of PTX, BF and CBG produced synergistic anticancer activity [32]. Gu et al. [33] found that the combination of BF and hydroxycampothecin improved the inhibitory effects of both drugs on castration-resistant prostate cancer in mice tumors compared with the administration of BF or hydroxycampothecin alone, potentially via the regulation of the PI3K/AKT/GSK-3β and p53-dependent apoptosis signaling pathways. According to Yuan et al. [34], the combination of trivalent arsenic derivatives (arsenite, As^III^) and CS-6 had synergistic cytotoxicity to human glioblastoma cell line U-87 cells. Each drug could induce G_2_/M cell cycle arrest in U-87 cells, which could be further enhanced by combination medication. CBL combined with gefitinib could suppress A549 cell viability and promote cell apoptosis by downregulating hepatocyte growth factor (HGF) protein expression and blocking cell-mesenchymal epidermal transformation factor (c-Met) gene amplification, which indicated that CBL might delay the occurrence of gefitinib resistance in lung cancer cells. The combined therapy might be a new, promising treatment for lung cancer patients whose cancer is resistant to gefitinib [35].

### 3.2. Cardiovascular Effects

Bufadienolides have two cardiovascular biological activities. One is the cardiotonic activity used to treat heart failure, the other is cardiotoxicity, which is a toxic side effect and can be used as part of the treatment to kill cancer cells, thereby showing anti-cancer activity. The side effect is that normal cells around the heart may also be killed [36]. Bufadienolides also have dual positive inotropic effects on cardiac muscle, and the cardiac mechanism is to directly inhibit the activity of α1Na^+^/K^+^-ATPase [37]. BF was shown to have a biphasic effect on the contractility of myocardial cells (hiPSC-CMs). At the initial stage of administration, the contraction of the myocardium was increased, conduction was accelerated, and the beating frequency increased. At the end of administration, the contraction of the myocardium was weakened, conduction disappeared and pulsation stopped. BF could reduce the action potential duration, cardiac action potential amplitude and maximum depolarization rate of hiPSC-CMs, and depolarize the resting membrane potential. Late sodium current enhancement is one of the main mechanisms for cardiotoxicity of BF [38].

Similar to digitalis, RBG has been shown to have both pharmacological and toxicological effects. High concentration of RBG could cause delayed depolarization and arrhythmia both in vitro and in vivo [39]. ABG inhibits the activity, migration, invasion and tubular formation of human umbilical vein endothelial cells (HUVECs) induced by vascular endothelial growth factor (VEGF) in vitro and inhibits the sprouting formation of VEGF-treated aortic rings. ABG blocks angiogenesis and interacts with the ATP binding site of VEGFR-2 through docking. ABG inhibits the autophosphorylation of VEGFR-2 induced by VEGF and inhibits the activity of the signal transduction pathway mediated by VEGFR-2. These findings indicate that ABG is a specific inhibitor of VEGF-mediated angiogenesis [40].

Ren et al. [41] revealed the synergistic mechanism of multiple components in Venenum Bufonis for ameliorating heart failure by integrating network pharmacology and molecular docking approaches. Their study identified four key targets (ATP1A1, GNAS, MAPK1 and PRKCA) and three potential active leading compounds (bufotalin, cinobufaginol and 19-oxy-bufalin) that may play a key role in myocardial contraction. The study showed that the anti-heart failure effect of toad venom was mainly related to the adrenergic signals involved in the process of myocardial contraction.

### 3.3. Anti-Inflammatory

Toad venom has been used for treating infectious and inflammatory diseases in China for thousands of years. Studies have shown that BF can reduce tumor necrosis factor-α (TNF-α)-induced interleukin-1beta (IL-1β), IL-6 and IL-8 production in rheumatoid arthritis fibroblast-like synoviocytes, suggesting an anti-inflammatory role of BF in these cells [42].

Asthma is a chronic inflammatory disease, and Th2 cells, such as IL-4, IL-5 and IL-13, play key roles in asthma pathobiology [43]. The present study showed that the NF-κB signaling pathway plays a vital role during the process of inflammation [44]. According to Zhakeer et al. [45], a mouse asthma model was developed by ovalbumin (OVA) sensitization and challenge in the BALB/c mice. In the mouse asthma model, BF significantly reduced hyperresponsiveness, strongly inhibited the increase of total inflammatory cells in BALF induced by OVA, significantly reduced the levels of IL-4, IL-5 and IL-13 in BALF and the specificity of OVA in serum IgE level, reduced inflammatory cell infiltration and goblet cell proliferation, inhibited IκBα degradation of NF-κB, and reduced the level of phosphorylated p65 protein in lung tissue. These data indicated that BF might exert its anti-inflammatory effects by inhibiting NF-κB activity.

Wang et al. [46] preliminarily proved that CBG could significantly reduce the number of M1 macrophages and pro-inflammatory cytokines, such as IL-6, TNF-α and inducible nitric oxide synthase (iNOS) expression in colitis induced by DSS. In addition, CBG also increased the number of M2 macrophages and the expression of anti-inflammatory factors in DSS-induced colitis in mice.

### 3.4. Analgesia

Toad venom can be used for analgesia, and bufadienolides monomers are the main components for their analgesia. Clinical studies showed that CBG had the advantage of a shorter onset time and longer duration time of analgesia, and it had a certain effect in the treatment of cancer-related pain [47]. CBG could prolong the paw withdrawal latency of mice in a hot plate experiment and inhibit the paw twisting response of mice in an acetic acid-induced mouse writhing experiment [48]. Nav channels (voltage-gated sodium channels, or VGSCs), which are transmembrane proteins responsible for generating and conducting action potential in excitable cells, are a key target for analgesia. BF could observably attenuate formalin-induced spontaneous leg withdrawal and licking responses as well as carrageenan-induced thermal and mechanical pain sensitivity in a dose-dependent manner, inhibiting the peak current of nav channels in dorsal root ganglion neuroblastoma (ND7–23 cells) in a dose-dependent manner [49].

### 3.5. Immunomodulating Activity

BF has been shown to directly balance the stimulation and inhibitory receptors on the surface of NK cells, indirectly activate natural killer (NK) cells by inhibiting the shedding of MICA, prevent immune escape, and indirectly enhance NKG2D-dependent immune surveillance. The results indicate that BF could directly or indirectly regulate immune response [50].

**Table 1 molecules-27-00051-t001:** Pharmacological activities of bufadienolides.

Pharmacological Action	Monomer	Dissolution/Doses	Effects/Mechanisms	Refs.
Antitumor	Bufalin	Dissolved in culture medium	Promoted proteasome activation and ATP1A1 protein degradation and thereby inhibited ATP1A1 expression in glioblastoma, inhibiting tumor growth and proliferation.	[20]
		Dissolved in DMEM	Induced endoplasmic reticulum stress via the IRE1-JNK pathway to inhibit the value added and promoted apoptosis of human hepatoma cell lines Huh-7 and HepG-2.	[51]
		Dissolved in dimethyl sulfoxide (maximum concentration 20 mg/mL)	Suppressed proliferation and induced apoptosis and G_2_/M phase arrest in pancreatic cancer cells.	[52,53]
		25, 50, 100 nM	Caused Annexin A2 and DRP1 oligomerization on the surface of mitochondria and disrupted the mitochondrial division/fusion balance to induce U251 cell apoptosis.	[54]
			Had marked antitumor activities by inducing apoptosis.	[55]
		Different concentrations	Reduced the phosphorylation of NOS3, thereby inhibiting the MAPK signaling pathway, and finally suppressed the gastric cancer peritoneal dissemination by inhibiting the EMT process.	[56]
		10 μM	Had an inhibitory effect on the growth and migration of ovarian cancer cells by inhibiting the activation of mTOR and the induction of HIF1α.	[57]
		Dissolved in DMSO, 1.0 mg/mL	Potentially acted on the Na^+^/K^+^ ATPase pump which is overexpressed in melanoma and had the highest anti-proliferative activity on melanoma cells.	[58]
		Diluted in DMSO, 80 nmol/L	Inhibited the proliferation of pancreatic cancer cells, and c-Myc downregulation enhanced this effect.	[59]
		0.1 mg/kg	Regulated cancer cell stem cells through CD133/NF-κB/MDR1 pathway to reverse colorectal cancer MDR.	[60]
	Cinobufagin	Injection (500 mg/mL)	Showed significant inhibition rates on gastric and hepatocellular tumor growth in vivo.	[61]
		50 μg/mL	Effective inhibition of breast cancer MDAMB-231 cell growth.	[62]
		0, 50, 100 nM	Induction of apoptosis in osteosarcoma cells via mitochondria-dependent intrinsic apoptotic pathway.	[63]
		Dissolved in 100% DMSO	Selectively suppressed cancer cell viability via DDR-mediated G_2_ arrest and apoptosis.	[64]
		0–500 nM	Reduced the proliferation and colony formation of human liver cancer cells in vitro, and induced mitotic arrest of human liver cancer cells.	[26]
	Cinobufotalin	0.1 μM and 0.2 μM	Inhibited de novo lipogenesis of hepatocellular carcinoma by binding SREBP1 to prevent SREBP1 from sterol regulatory elements and decreasing SREBP1 expression.	[65]
	Arenobufagin	Formulated in physiological saline	Inhibited the proliferation of SW1990 and BxPC3 cells and induced cell arrest, apoptosis and autophagy.	[66]
		Dissolved in DMSO (10 mM)	Inhibited the proliferation and survival of HER2 overexpressing breast cancer cells.	[30]
		Dissolved in DMSO to a concentration of 500 µM	Showed potent antineoplastic activity against HCC HepG2 cells and corresponding multidrug-resistant HepG2/ADM cells.	[67]
		10, 20, 40, 100, 150 and 200 ng/mL	Downregulated the expression levels of Cdc25C and Cyclin B1, reduced the cell survival rate.	[22]
		Dissolved in DMSO	Showed selective tumor killing effect on refractory cancer cells.	[23]
	1β-OH-ABG	At different concentrations	Significantly reduced the expression levels of p-AKT/AKT and p-mTOR (Ser2248 and Ser2481)/mTOR in a time-dependent manner.	[24]
	Resibufogenin	Dissolved in DMSO (0.1%)	Inhibited proliferation, migration and invasion of ovarian clear cell carcinomas (OCCC), and induced apoptosis in them.	[68]
	Hellebrigenin	10, 20, 40, 100, 150 and 200 ng/mL	Distinct cytotoxicity against cancerous glial cells with high potency and selectivity.	[22]
		48 nM in SW1990 and 15 nM in BxPC-3	Inhibited pancreatic cancer cells’ proliferation by inducing cell apoptosis and activation of autophagy via upregulation of apoptosis-related proteins and the autophagic key proteins.	[69]
		Dissolved in DMSO	Induced apoptosis and induced G_2_/M cell cycle arrest.	[23]
	Marinobufagin Telocinobufagin	200 mg/mL	Marinobufagin and telocinobufagin have shown remarkable biological action on hematological, solid, sensitive and/or resistant human tumor cell lines.	[25]
	ψ-bufarenogin	Dissolved in DMSO	Inhibited the proliferation of liver cancer cells by blocking the cell cycle transition, and downregulated the expression of Mcl-1 to promote cell apoptosis.	[28]
	19-Hydroxybufalin	10 mM in DMSO	Inhibit the proliferation, migration and invasion of NSCLC cells and promoted the apoptosis of NSCLC cells through the Wnt/β-catenin pathway.	[29]
Cardiovascular effects	Bufalin	100 mmol/L	Had a biphasic effect on cardiomyocyte contractility.	[38]
		10 nM	Suppressed tumor microenvironment-mediated angiogenesis by inhibiting the STAT3 signaling pathway in vascular endothelial cells.	[70]
	Marinobufagin	0.025, 0.05, and 0.1 nmol·min^−1^·g body wt^−1^	Interacted with the ouabain binding site of the α1Na^+^-K^+^-ATPase subunit and thereby influenced cardiac inotropy.	[71]
			As a biomarker for preeclampsia.	[72]
	Resibufogenin	0.2 mg/kg, iv	Induced delayed afterdepolarization and triggered arrhythmias both in cardiac fiber in vitro and in beating heart in vivo at high concentrations.	[39]
		0.3, 1, 3, 10, and 30 μM	Exhibited promising antitumor effect through antiangiogenesis in vivo without obvious toxicity.	[73]
		1–100 μM	Influenced the cardiac electrical conduction by its multi-channel blocking actions and possessed a proarrhythmic effect at a lower concentration in the working heart of guinea pigs.	[74]
	Arenobufagin	Dissolved in DMSO	Inhibited vascular endothelial growth factor (VEGF)-induced viability, migration, invasion and tube formation in human umbilical vein endothelial cells (HUVECs) in vitro.	[40]
Anti-inflammatory	Bufalin	100 μL of serum-free medium containing 0, 10, 20, and 30 μM	Suppressed inflammatory cell increase.	[42]
	Cinobufagin	50 mg/mL	Significantly decreased the number of proinflammatory factors.	[46]
	Gammabufotalin	At nontoxic doses	Inhibition of NF-κB activity exerted anti-inflammatory effects.	[44]
Analgesia	Bufalin	1, 5 or 20 µM	Inhibited the peak current of nav channels that generate and conduct action potentials in excitable cells.	[49]
	Cinobufagin	2 g were soaked in 10× volume of water for injection	Showed stronger analgesic activity and less hepatotoxicity.	[48]
Immunomodulating activity	Bufalin	20, 50 nM; 50, 100 nM	Directly or indirectly regulated immune response.	[50]

## 4. Strategies to Improve Solubility and Bioavailability

Structural modification and pharmaceutical formulation strategies are the methods to improve the solubility and bioavailability of bufadienolides. Structural modification includes the preparation of derivatives and prerequisite drugs, and pharmaceutical formulation strategies include solid dispersions, cyclodextrin inclusion complexes and nanodrug delivery systems, and so on (Figure 4).

### 4.1. Structural Alterations

Structural modification can change the performance of drugs and is one of the effective ways to develop new drugs. Structural modification has succeeded in expanding the antimicrobial spectrum, enhancing antimicrobial activity, overcoming drug resistance, improving pharmacokinetic properties, reducing toxic side effects and adapting to formulation needs [75,76,77]. The structure–activity relationship of bufadienolides revealed by 3D-QSAR study is shown in Figure 5, which provides reliable information for structure modification and rational drug design of bufadienolides with anticancer activities in medical chemistry [78].

#### 4.1.1. Derivative

Preparation of derivatives is a widely used approach to improve the solubility, bioavailability and pharmacological effects of bufadienolides. BF211 (Figure 6a) is a derivative of BF, and exhibited significantly improved solubility (increased from 10 μg/mL to 2500 μg/mL) and stronger cytotoxicity against multiple myeloma than BF [14]. According to Wu et al. [79], BF211 had a stronger inhibition effect on 20 cancer cell lines and lower acute toxicity at nanomolar concentrations compared with BF, and could inhibit the proliferation of A549 cancer cells and induce apoptosis by activating the signal cascade downstream of the Na^+^/K^+^-ATPase signal body [80].

Bufalin-3-yl nitrogen-containing-carbamate derivatives 3 (Figure 6b) had proliferation inhibition activity on human cervical epithelial adenocarcinoma (HeLa) cell line [14]. Bufalin-3-piperidinyl-4-carboxylate compound (Figure 6c) had significant cytotoxicity compared with BF [81].

The structure–activity relationship indicated that the substituents at position C-16 were the key factor affecting the activity of cinobufagin-3-yl nitrogen-containing-carbamate derivatives (Figure 6d), and the variation trend was as follows: acetic ester » benzoic ester ≈ hydroxy > carbamate. The nitrogen-containing side chain effectively improved the solubility of the compounds, so that the synthetic carbamate derivatives containing cinobufagin-3-yl nitrogen showed significant in vitro antiproliferative activity against cancer cell lines [82]. Chen et al. [83] designed and synthesized ten 3-monopeptide substituted ABG derivatives. The result indicated that monopeptide substitutions on 3-hydroxy of ABG could improve their antiproliferative activity, and among them, the compound ZM226 (Figure 6e) was a potent antitumor agent with low cardiotoxicity. These findings suggested that optimizing ABG at position 3 might be an effective strategy for the development of antitumor drug candidates derived from ABG.

#### 4.1.2. Prodrug

Structural modifications also employ biosynthesis by adding precursors. Prodrugs are common methods for chemical structure modification of nervous system drugs, antitumor system drugs and antiviral drugs, which can increase the solubility and bioavailability of drugs, enhance targeting, and reduce the toxicity and side effects of drugs. Bufalin 3-phosphate (Figure 7a) is a water-soluble prodrug that could solve the problem of water solubility of BF and facilitate its administration in vivo. Furthermore, bufalin 3-phosphate was also demonstrated to effectively inhibit tumor growth in an orthotopic triple negative breast cancer (TNBC) mouse model [84]. Liu et al. [85] obtained bufalin-3-*O*-β-d-glucopyranoside (**3**) (Figure 7b) and bufalin-3-*O*-[β-d-glucopyranosyl (12)-β-d-glucopyranoside)] (**4**) (Figure 7c) by using an efficient chemical enzymatic method to glycosylate bufalin. The water solubility of compounds (**3**) and (**4**) was 13.1 and 53.7 times that of BF, respectively. PEGS-BF (Figure 7d) was a novel polyethylene glycol (PEG)-based polymeric prodrug of BF, which could improve its water solubility and stability on the premise of maintaining its original anticancer activity. The results showed that the solubility of PEGS-BF was about 17.3 mg/mL, which was about 21 times higher than that of free BF (32.76 μg/mL) [86].

Liu et al. prepared a new octreotide (oct)-modified esterase-sensitive tumor-targeting bufalin polymer prodrug, which was constructed by the polymer prodrugs of hydrophobic small molecule drugs and had good water solubility. The introduction of tumor-targeting moieties on the polymer prodrugs could effectively improve the accumulation of drugs in tumor tissues and the therapeutic effects of drugs. The experiments in vivo and in vitro had proved that P(OEGMA-co-BF-co-Oct) (Figure 7e) had stronger anti-cancer activity than free BF [87]. The prodrug of bufalin, P(OEGMA-co-BSTMA)-g-P(DEA-co-BMA) (Figure 7f), obtained by combining bufalin with tumor-targeting peptide (RGD) and endosomal escape polymer poly (*N*,*N*-diethylaminoethyl methacrylate-butyl methacrylate copolymer) (P(DEA-co-BMA)), showed better anticancer effect [88].

Deng et al. [89]. synthesized a fibroblast activation protein α (FAPα)-activated arenobufagin prodrug 3f (Figure 7g). The results showed that prodrug 3f significantly improved tumor targeting, reduced cardiac toxicity and enhanced antitumor activity. Chai et al. [90] used a FAPα-based prodrug strategy to synthesize a dipeptide (Z-Gly-Pro)-conjugated BF211 prodrug named BF211-03 (Figure 7h). Compared to BF211, BF211-03 showed equal anticancer activity and lower weight loss in mice bearing HCT-116 xenografts.

In short, through the structural modification method of preparing bufadienolides derivatives and prodrugs, the solubility and bioavailability of bufadienolides can be effectively improved under the premise of maintaining or enhancing their original biological activities.

### 4.2. Solid Dispersion

Solid dispersion is a highly dispersed mixture of drug and carrier. The solid dispersion technique can solve the problems of poor water solubility for poorly water-soluble drugs. Zou et al. prepared a new type of spray-solidified solid dispersion, which realized the rapid simultaneous dissolution of three therapeutic auxiliary drugs: BF, CBG and RBG. The results showed that all drugs were dispersed in the matrix in molecular form, and the dissolution rate was significantly improved (about four times). Furthermore, synchronized release of different drugs from a single carrier was achieved due to the highly molecular dispersibility and the excellent solubilization properties of F127 [91].

### 4.3. Cyclodextrin Inclusion Complexes

Cyclodextrin (CD) is composed of 6–12 D-glucose molecules linked by 1,4-glycosidic linkages as cyclic oligosaccharides, with β-CD being the most common. Both ends and the outside of β-CD are hydrophilic, while the inside of the cartridge is hydrophobic. Using cyclodextrin inclusion technology can increase the stability, solubility and bioavailability of drugs. Guo et al. proved that the ball-milling method to prepare RBG β-CD inclusion compounds was the best preparation method. The solubility and in vitro dissolution of RBG were significantly improved after β-CD inclusion, and the irritation was significantly reduced [92]. After BF was encapsulated by β-CD, the solubility in water and phosphate buffer solution (pH = 7.4) increased by 24 times and 34 times, respectively. In addition, a conjugate of β-CD and folic acid (FA) was prepared, and the inclusion compound and FA-targeting inclusion compound had better dissolution and solubility than pure BF inclusion compound. The HCT116 cytotoxicity test showed that the antitumor effect of BF in the presence of β-CD and FA-conjugated β-CD (FA-β-CD) more than doubled, indicating that β-CD (FA-PEI-β-CD) (Figure 8) might be used as a carrier of BF for antitumor therapy [93].

### 4.4. Nanodelivery Strategies

Nanomedicine represents a favorable tool to increase bioavailability and activities of natural products. Nanodrug delivery systems usually refer to dispersions with diameters of 1–1000 nm as carriers [94]. Nanocarriers are usually developed from polymers, lipids/proteins, metals and carbon-based materials. Compared with traditional drug carriers, nanodrug carriers can solve the pharmacological limitations of poor drug resistance, poor bioavailability and poor water solubility [95,96]. Generally, nanocarriers provide a large surface area and can overcome anatomical obstacles [97]. A nano-targeted formulation can also use a specific drug carrier or drug delivery technology to clearly focus the drug on a specific tissue or organ. The nano-targeted formulation not only has the characteristics of targeting specificity, but also increases the solubility of hydrophobic drugs and increases the stability of drugs [98,99]. At present, there are several nanodelivery strategies for bufadienolides, including nanoparticles, nanoliposomes, polymeric micelles, microemulsion, dendrimer and nanosuspension. The nanodelivery strategies of bufadienolides are shown in Table 2.

#### 4.4.1. Nanoparticles

Nanoparticles as a drug delivery system have been extensively studied over the past twenty years. Nanoparticles can improve the solubility and the stability of encapsulated cargos, and even transfer across biological barriers such as the intestinal tract or blood–brain barrier [100]. Many studies have demonstrated the ability of nanoparticles to offer significant therapeutic improvement [101]. The development of nanoparticle delivery has been extended to the delivery of chemotherapeutic drugs to obtain better curative effects while reducing the toxicity of cytotoxic drugs [102,103,104,105]. According to Xu et al. [106], epidermal growth factor (EGF)-modified nanospheres with encapsulated BF increased toxicity in colorectal cancer cells. BF loaded in calcium phosphate/DPPE-PEG-EGF hybrid porous nanospheres had strong targeting and controlled release capabilities, could accurately deliver the drug to tumor cells, effectively increased the tumor local drug concentration, and improved the stability of BF and absorption rate to improve its efficacy. The pluronic polyetherimide nanoparticles loaded with BF had slow release and tumor-targeting effects, and significantly inhibited the growth and metastasis of colorectal cancer [107].

Compared with BF, bufalin-bovine serum albumin nanoparticles (BF-BSA-NPs) had significantly increased half-life and good solubility, and could increase the bioavailability of BF by prolonging the systemic circulation time and increasing the amount of BF available for tissue uptake. Furthermore, BF-BSA-NPs had a high liver uptake rate and strong anti-liver cancer activity, and was a promising liver-targeted drug delivery system [108]. Zhang et al. found that the blood plasma area under curve, the mean retention time and the terminal half-life of BF-loaded bovine serum albumin nanoparticles were 1.19 to 1.81, 2.12 to 3.61 and 2.17 to 2.94 times that of BF, respectively. Therefore, BF-BSA NPs had the function of sustained release, thus prolonging the BF remaining in the blood [109]. Yin et al. found that mPEG-PLGA-PLL-cRGD nanoparticles (BNP) loaded with BF could effectively target tumors in a SW620 xenograft mouse model, and were significantly more effective than other NPs in inhibiting tumor growth. The reason was that BNP had good stability, and the water-soluble mPEG block could produce nanoparticles with good biocompatibility and long-cycle characteristics, thereby enhancing the targeting ability of nanoparticles [110].

According to Xu et al. [111], a novel albumin polymer hybrid with a core–shell structure was designed to target delivery of BF. The sheath layer was composed of ursodeoxycholic acid (UA)-modified bovine serum albumin (UA-BSA), while the stable core consisted of poly n-butyl cyanoacrylate (PBCA) nanoparticles. This novel designed hybrid albumin nanocomplex could effectively improve the therapeutic effect as well as reduce toxicity, including hemolysis, vascular irritation and cardiotoxicity, and would be a potential carrier for anticancer drug.

Fan et al. developed a biomimetic nanomedicine system (GTDC@M-R NPs) for triple-negative breast cancer (TNBC) treatment. The nano system had good biocompatibility, good tumor tissue-targeting accumulation ability and longer blood circulation time, and it could inhibit the growth and metastasis of TNBC and effectively improve the cell and nuclear-targeting ability of nano systems [112]. Dong et al. prepared RBG-Gal-SP188-PLGA nanoparticles (RGPPNs) which could improve the solubility of RBG. The study of mouse liver cancer models showed that RGPPNs had better active liver targeting [113]. Chu et al. prepared RBG-loaded poly(lactic-co-glycolic acid) (PLGA)-d-α-tocopherol polyethylene glycol 1000 succinate (TPGS) nanoparticles (RPTN) to enhance the treatment of liver cancer. TPGS could be used as a solubilizer or enhancer to improve bioavailability by increasing the water solubility of hydrophobic drugs and reducing P-glycoprotein (P-gp)-mediated multidrug resistance of cancer cells. RPTN could enhance the pharmacological effects of liver targeting and reduce the toxicity of RBG [114]. Li et al. designed a targeted delivery and smart responsive dopamine (PDA)-based nanomedicine for the delivery of CBG to treat lung cancer. This nanocarrier was not only sensitive to the low biological pH value of on-demand drug release, but also biodegradable and decomposed into biocompatible end products. Therefore, the polydopamine nanodrug delivery system modified by CBG loaded with FA, as a multifunctional nanoplatform, had the potential to improve bioavailability and reduce the side effects of chemotherapeutics [115].

Liu et al. constructed a nanosphere system called HA@RBC@PB@CS-6 NPs (HRPC) (Figure 9). In this system, Prussian blue nanoparticles (PB NPs) with hollow porous structure were used as the carrier for CS-6 and a photothermal sensitizer simultaneously. After the erythrocyte membrane was wrapped on the surface of the PB nanoparticles, the blood circulation life was extended to 10 h, and the immune escape ability was increased by more than 60%, which was conducive to the high concentration of HRPCs aggregation of the targeting molecule hyaluronic acid (HA) at the tumor site [116]. Xiao et al. [117] developed a near-infrared and redox-responsive nanocomplex PCDI@M (Prussian blue@CS-6@polydopamine-indomethacin@M, Figure 10) loaded with CS-6 and indomethacin nanocomplexes for enhancement of tumor suppression and reprogramming of the inflammatory microenvironment. The designed nanocomplexes had high biocompatibility and enrichment at the tumor site. The combination of photothermal therapy and chemotherapy could reduce the side effects of CS-6 on normal cells. In vitro studies had shown that redox-induced indomethacin release could effectively reduce tumor tissue inflammation caused by photothermal therapy. More importantly, the presence of indomethacin increased the sensitivity of tumor cells to CS-6 by strongly reducing the secretion of prostaglandin E2. In vivo studies had shown that laser-assisted nanocomplexes had excellent tumor suppressive properties.

In summary, nanoparticles loaded with bufadienolides could not only improve the solubility and bioavailability of bufadienolides, but also achieve sustained-release and targeted drug delivery effects, thereby improving its antitumor activity.

#### 4.4.2. Nanoliposomes

Nanoliposomes are a new type of nanodrug delivery system, which have the characteristics of high nanoparticle physical stability, less drug leakage, good sustained release, low toxicity and cell-specific targeting [118,119,120]. Lipid nanoparticles can deliver a sustained and high level of the loaded drug in the blood plasma [121], increase the therapeutic index of many drugs, and provide drug targeting and controlled release functions [122]. Li et al. prepared a bufadienolides liposome (BU-lipo). The solubility of BF, CBG and RBG in aqueous medium under different pH conditions was 20–60 μg/mL. The stability of BU-Lipo freeze-dried formulation containing 10% trehalose in a desiccator at 2–8 °C could be up to 6 months [123]. Nanostructured lipid carriers containing bufadienolide (BU-NLC) could be used in gastrointestinal applications. The incorporation of bufadienolides into NLC reduced the side effects of bufadienolides solution, and showed excellent antitumor effects and good blood compatibility [124]. Hu et al. added bufadienolides whose main components were BF, CBG and RBG into poloxamer-modified liposomes. Compared with the bufadienolides solution and unmodified liposomes, the bufadienolides liposomes significantly prolonged the retention time of bufadienolides in plasma, and increased the AUC of bufadienolides, thereby improving the antitumor effect [125]. In order to improve the release efficiency of the derivative BF211 liposome, Gao et al. developed a surfactant-assisted rapid release strategy, and reduced side effects by reducing non-specific biodistribution, which effectively addressed the poor in vivo targeting and serious toxic side effects of BF211 [126]. According to Yuan et al. [127], the drug release rate of BF-loaded PEGylated liposomes was slower than bufalin-loaded liposomes. The pegylated liposomes loaded with BF were significantly more toxic to U251 cells compared with BF entities, and could prolong or eliminate the half-life of BF in rat plasma, which indicated that BF-loaded PEGylated liposomes could increase the solubility in plasma and increase the drug concentration. Chen et al. prepared BF liposomes co-modified with transferrin (Tf) and FA ((FA + Tf) BF-LPs) (Figure 11A) for the treatment of lung cancer. Experiments showed that the release rate of BF in (FA + Tf) BF-LPs slowed down and had a burst effect in the early stage. (FA + Tf) BF-LPs could effectively inhibit the proliferation of A549 cells, and had obvious subcutaneous tumor targeting and the potential to actively deliver drugs to tumor tissues. This study showed that BF co-modified with TF and FA was a promising lung targeting agent [128].

Liu et al. developed a wheat germ agglutinin (WGA)-grafted lipid nanoparticles of BF, which could overcome the disadvantage of low solubility of BF by oral administration. Studies on the correlation between 6-coumarin fluorescently grafted WGA lipid nanoparticles and Caco-2 monolayer showed that WGA enhanced the uptake of nanoparticles by cells compared with lipid nanoparticles without WGA [129]. Compared with suspension, the adhesion of WGA-grafted lipid nanoparticles of BF to the intestinal mucosa was significantly enhanced (*p* < 0.05), and WGA-grafted lipid nanoparticles of BF showed greater AUC and C_max_, and oral bioavailability was increased by 2.7 times [130]. These results illustrated the potential use of WGA-grafted lipid nanoparticles in oral administration of poorly soluble drugs such as BF.

#### 4.4.3. Polymeric Micelles

Polymer micelles are a thermodynamically stable colloidal solution formed by self-assembly of synthetic amphiphilic block copolymers in water. As a drug carrier, polymer micelles could increase the solubility and bioavailability of hydrophobic drugs, and the design of the hydrophilic shell gave a long circulation effect. VES was a water-soluble derivative of natural vitamin E, which could effectively encapsulate insoluble drugs. Yuan et al. prepared vitamin E succinate grafted oligosaccharides (VES-CSO) and a cyclic (arginine-glycine-aspartic acid) (RGD) modified TPGS multifunctional drug delivery system (Figure 11B) to improve the efficacy of drug-resistant colon cancer. The results showed that BF-loaded VES-CSO/TPGS-RGD mixed micelles (BF@VEC/T-RGD MM) had higher treatment efficiency and fewer side effects compared with free BF, as well as tumor targeting due to a stronger penetration and retention effect [131,132]. Shi et al. prepared a polymeric prodrug with BF, a tumor-targeting polypeptide (RGD), and endosomal escape polymer poly (*N*,*N*-diethylaminoethyl methacrylate-co-butyl methacrylate) (P(DEA-co-BMA)) as carriers (BUF-NP-RGD, Figure 11C). The results of cell survival rate and tumor-bearing nude mice model indicated that BF-NP-RGD had better anticancer performance in vitro and in vivo compared with free BF. The reason was that PEG-BF, on the premise of maintaining the original in vitro and in vivo anticancer activity, exhibited significantly improved aqueous solubility and stability, and the multifunctional polymeric prodrug after covalent attachment with RGD exhibited better anticancer performance in addition to improved aqueous solubility and stability [88].

Yuan et al. prepared ABG-loaded PN (ABG-PN) (Figure 11D), which promoted the systemic release by significantly increasing the solubility of ABG. ABG-PNS could improve drug pharmacokinetics, the area under the curve increased by 1.73 times, and the elimination clearance rate decreased by 37.8%. In addition, ABG-PN could enhance the anti-cancer effect of pure drugs by increasing the uptake of drug molecules by cells [133]. Gou et al. [134] synthesized the corresponding dual-targeting immunomicelles (DTIs) loaded with bufalin (DTIs-BF) by copolymer self-assembly in an aqueous solution. Compared with free BF, DTIs-BF significantly enhanced the internalization and cytotoxicity of SMMC-7721 cells in vitro. In addition, the therapeutic effect on SMMC-7721 cells was further enhanced by the programmed cell death specifically caused by bufalin.

#### 4.4.4. Microemulsion

##### Submicroemulsion

Submicroemulsion is a colloidal dispersion system with liquid fat oil as the core, phospholipids as the main emulsifier, and the droplet size of submicroemulsion is between emulsion and microemulsion. It has the characteristics of good stability, good antitumor effect and low toxicity [135]. In addition, by incorporating the drugs into the oil phase and interfacial layer, this delivery system could reduce irritation and toxicity, increase solubility and stability, as well as offer the possibility of sustained release. Li et al. prepared bufadienolide-loaded oral submicron emulsion (BU-OE) by high-pressure homogenization, and the mean particle size of BU-OE was 142.2 ± 52.6 nm. BU-OE was an organic combination of three bufadienolides (BF, CBL and RBG), with higher purity, lower toxicity, wider antitumor spectrum and better curative effect [136].

##### Microemulsion

Microemulsion is a transparent or translucent, low-viscosity, isotropic and thermodynamically stable oil–water mixture system spontaneously formed by an aqueous phase, an oil phase and a surfactant at an appropriate ratio. Microemulsion is a new ideal drug delivery carrier, which has the characteristics of stable absorption, targeted release, improving efficacy and reducing side effects. The droplet diameter range of microemulsion is 10–100 nm [137,138]. Liu et al. prepared an improved self-microemulsifying drug delivery system (SMEDDS) of BF with droplet size of 33.9 nm, which was well absorbed in all intestinal segments. In terms of relative bioavailability, the absorption of BF in SMEDDS was 2.38 times higher than that of BF suspension, the equilibrium solubility was 12.6 mg/mL and the soluble drug post-digestion was 73.6%. The accumulated amount of drug released from the SMEDDS in 120 min was approximately 90%, as opposed to 35% from suspension. The results showed that SMEDDS released faster than suspension [139].

Li et al. encapsulated BF in a low-viscosity microemulsion, and the release of BF occurred in the lamellar liquid crystal structure produced in situ. Cytotoxicity, apoptosis and pharmacokinetic experiments showed that the bioavailability of BF after encapsulation was significantly improved. The study exploited the structural features of the in situ phase transition from a microemulsion to a lyotropic liquid crystal toward the aim of controlled release of BF-loaded carriers [140].

#### 4.4.5. Dendrimer

Dendrimers are highly symmetric spherical compounds with good encapsulation properties and can be used to deliver hydrophobic compounds and anticancer drugs [141,142,143,144,145]. In order to improve the bioavailability of BF, Chan et al. embedded BF in a novel polypeptide-dendrimer (PD) to obtain a BF-peptide-dendrimer inclusion compound (BPDI) (Figure 12). Small molecules with poor water solubility could be attached to dendrimers through physical inclusion or non-covalent bonding to improve water solubility and intestinal absorption. The results of the in vitro Caco-2 cell monolayer model showed that the Papp value of BF encapsulated in the form of BPDI was about three times higher than that of free BF, which indicated that the intestinal permeability of BF could be improved [146].

#### 4.4.6. Nanosuspension

Nanosuspension is a colloidal dispersion of nanodrug particles prepared by various methods, which is suitable for all drugs insoluble in water and oil. It not only solves the problem of poor solubility and bioavailability of drugs, but also changes the pharmacokinetics of drugs, thereby improving the safety and effectiveness of drugs [147,148]. Zou et al. developed a multicomponent nanosuspension formulation of bufadienolides using a wet-milling technique to improve their dissolution behavior. The results showed that the dissolution performance and stability of BF, CBG and RBG were significantly improved, and the use of wet milling to manufacture nanosuspensions was a promising method to achieve rapid and simultaneous dissolution of multi-component preparations [149].

#### 4.4.7. Sub-Microspheres

Sub-microspheres have good water solubility and blood compatibility, and are expected to be candidates for oral administration [150,151]. Xu et al. [152] developed multifunctional albumin sub-microspheres to co-deliver BF and nintedanib for tumor-targeted combination therapy with a particle size of 879 ± 56 nm. The biguanide and ursodeoxycholic acid dual-modified multifunctional albumin were synthesized to enhance the antitumor effect and tumor-targeting efficiency. In vitro and in vivo experiments demonstrated that the multifunctional albumin sub-microspheres possessed superior tumor-targeting efficiency. Furthermore, nintedanib and BF combined therapy relieved the tumor microenvironment and exerted a synergistic therapeutic effect. 

**Table 2 molecules-27-00051-t002:** Nanodrug delivery systems.

	Materials	Experimental Subject	Properties	Ref.
Nanoparticles	Bufalin-loaded CaP/DPPE-PEG-EGF	HCT-116 cells, Male BALB/c mice	Showed improved antitumor effects on colon cancer in nude mice, but without severe side effects.	[106]
	Bufalin-loaded pluronic polyetherimide	HCT116 cells, male athymic nude mice	Had controlled release effect, protected normal tissues from bufalin injury during blood circulation and realized directional and controlled release.	[107]
	Bufalin-loaded bovine serum albumin	Kunming mice, nude mice SMMC-7721 cells	Had higher liver uptake and stronger antitumor activity against hepatocellular carcinoma.	[108]
		Male Wistar rat	Reduced side effects to a certain extent.	[109]
	Bufalin-loaded mPEG-PLGA-PLL-cRGD	SW620 colon cancer cells, and BALB/c female athymic nude mice	Had good stability, sustained release and tumor targeting, and sustained release in vitro for more than 192 h.	[110]
	Bufalin-loaded albumin–polymer hybrid	HepG2 cells, male SD rats	Had good stability, effective tumor targeted delivery potential and side effects reduction ability.	[111]
	Gamabufotalin-loaded RBC membrane camouflaged Prussian blue	MDA-MB-231 cells, tumor-bearing BALB/c mice	Prominent in synergistic photothermal/chemotherapy for tumors without side effects on normal tissues.	[116]
	Gamabufotalin-loaded GTDC@M-R	MDA-MB-231 cells, BALB/c mice	Showed long blood circulation time, improved bio-safety and accurately accumulated at the tumor site.	[112]
	RBG-loaded Gal-SP188-PLGA nanoparticles	HepG2 cells, Kunming mice	Showed excellent in vivo therapeutic effects and anticancer effects, and reduced toxicity.	[113]
	Resibufogenin-loaded poly (Lactic-co-glycolic acid)-d-α-tocopheryl polyethylene glycol 1000 succinate	HepG2 cells, Kunming mice	Enhanced the pharmacological effects of liver targeting and reduced the toxicity of RBG.	[114]
	Cinobufagin-loaded and folic acid-modified polydopamine	Beas2B, A549, and LLC cell lines, male nude mice	Better therapeutic effect on lung cancer when combined with photothermal therapy.	[115]
	Biomimetic nanoparticles loading with gamabufotalin-indomethacin	RAW 264.7 cells, Hela cells, BALB/c mice	Had high biocompatibility and enrichment at the tumor site and reduced side effects of CS-6 on normal cells.	[117]
Nanoliposomes	Liposome-encapsulated BF, CBG and RBG		Improved stability.	[123]
		Wistar rats, Kunming mice, HGC-27 and U87-MG cell lines	Reduced side effects, and showed excellent antitumor effects and good blood compatibility.	[124]
		Lovo cells, NCI-H157 cells, SD rats and Kunming mice	Had slow-release properties and better antitumor effect and safety.	[125]
	PEGylated BF211 liposomes	HepG2 cells, BALB/c mice and BALB/c nude mice, pigmented guinea pig, and SD rats	Prolonged blood circulation time, reduced cardiac toxicity, improved tolerance and improved the drug properties.	[126]
	Bufalin-loaded PEGylated liposomes	Male SD rats, U251and U87 cells	Improved the solubility and increased the blood concentration of the drug.	[127]
	Bufalin liposomes co-modified with transferrin and FA	A549 cells, male BALB/c nude mice	Had the potential to actively deliver drugs to tumor tissues, inhibited tumor growth in mice and had no systemic toxicity.	[128]
	Bufalin-loaded wheat germ agglutinin-grafted lipid	Caco-2 cells	Enhanced cell uptake of nanoparticles.	[129]
			Showed greater AUC and C_max_, increased oral bioavailability-by 2.7 times.	[130]
Polymeric Micelles	Bufalin-loaded vitamin E succinate-grafted-chitosan oligosaccharide/rGd conjugated tPGS mixed micelles	HCT116 cells, male SD rats, BALB/c-nu/nu	Exhibited good stability, sustained-release pattern, higher intracellular uptake and greater cytotoxicity.	[131]
	Bufalin-loaded endosome-escaping polymer and tumor-targeting peptide	HCT116 cells, male SD rats, female BALB/c	Had a better anticancer effect, promoted cell apoptosis and inhibited angiogenesis and anti-proliferation.	[88]
	ABG-loaded polymeric nanomicelles	HepG2 cells, male SD rats	Increased the cellular uptake of drug molecules to enhance the anticancer effect of pure drugs.	[133]
	DTIs loaded with BF (DTIs-BF)	SMMC-7721 cells	Enhanced the internalization and cytotoxicity of SMMC-7721 cells, and further enhanced the therapeutic effect on SMMC-7721 cells.	[134]
Microemulsion	BF, CBG and RBG-loaded submicron emulsion	ICR mice, BALB/c-nu nude mice, SD rats, HepG2, HCT-8, BGC-803, and EC9706 cells	Had a significant inhibitory effect on HepG2, HCT-8 and EC9706 cells, a slight inhibitory effect on BGC-803 cells in nude mice and no obvious toxicity to mice.	[136]
	A bufalin self-microemulsifying drug delivery system	Male SD rats	Significantly improved solubility and bioavailability and well absorbed in all intestines.	[139]
	Liquid crystalline carriers of bufalin	A549 cells, Wistar rats	1.4 times enhancement of the cytotoxicity in comparison to the pure BF suspension, increased bioavailability.	[140]
Dendrimer	Bufalin-peptide-dendrimer inclusion through Caco-2 cell monolayer	Caco-2 cells	Improved intestinal permeability and bioavailability.	[146]
Nanosuspension	Multicomponent amorphous BF, CBG and RBG nanosuspension		Improved the dissolution performance, and realized the rapid and simultaneous dissolution of multi-component preparations.	[149]
Sub-microspheres	Co-delivery of bufalin and nintedanib via albumin sub-microspheres	HepG2 cellse, male ICR mice, H22 cells	A core-shell structure that enables payload efficiency and stability, good tumor targeting properties, alleviated the tumor microenvironment, exerted a synergistic therapeutic effect.	[152]

## 5. Conclusions

Bufadienolides have various pharmacological activities, among which the antitumor effect of bufadienolides has attracted considerable attentions in recent years. At present, the anticancer research on bufadienolides is mainly focused on several active ingredients, such as bufalin, cinobufagin, arenobufagin and resibufogenin. However, 96 varieties of bufadienolides have been isolated and identified in toad venom, so it needs to be explored whether other compounds are active. Moreover, the low water solubility and bioavailability of bufadienolides limit their clinical application. Structural modification and pharmaceutical preparations have great potential in improving solubility and bioavailability of bufadienolides. Structural modification is a way to improve drug solubility based on the structure–activity relationship of bufadienolides. The strategy of pharmaceutical approaches is to improve the solubility assisted by solid dispersion, microemulsion, submicroemulsion, cyclodextrin inclusion and nanodrug delivery systems. Among these approaches, nanodrug delivery systems represent an ideal drug loading system with great potential for tumor treatment. In addition, the side effects of bufadienolides, such as hemolysis, vascular irritation, and cardiotoxicity, were remarkably attenuated by being loaded into the nano-targeted carrier. Although structural modification and pharmaceutical formulations strategies have improved the solubility and bioavailability of bufadienolides, these methods have their own limitations in clinical applications. For example, the instability problems of liposomes, such as aggregation, fusion and drug leakage during storage, have not been resolved. Furthermore, some nanoparticles and polymer micelles are only prepared at laboratory scale, and the action mechanism of some materials still need further study.

According to studies, most of the anticancer research and administration strategies on toad venom have focused on BF, which is the most abundant bufadienolide monomer, but some studies showed that the other ingredients in bufadienolides also have strong pharmacological effects. The coexistence of multiple components in traditional Chinese medicine may help the pharmacological effects to play a synergistic effect. A new strategy for overcoming the poor solubility and increasing in vivo efficacy of drug is to encapsulate several components in nanodrug delivery systems, which would increase their therapeutic effects and in vivo release characteristics. Furthermore, the co-loaded nano-formulations can exert the synergistic therapeutic effect of multiple components. This strategy has been used for loading multiple traditional Chinese medicine components [153,154], and can be used for bufadienolides. However, before co-loading, it is necessary to conduct detailed research on the physical and chemical properties and pharmacological effects of each component and the pharmacological effects after co-loading. At present, some beneficial explorations have been carried out in bufadienolides, such as multiple components co-loading in solid dispersion and submicroemulsion. Additionally, for bufadienolides and chemotherapeutic drugs, a nanodrug combinational strategy for efficient cancer therapy with intrinsic tumor microenvironment-responsive elements and low side effects is highly desired. However, it is still in the preliminary stage, and the balanced encapsulation of multiple components, release characteristics and synergistic pharmacological effects still need further study.

## Figures and Tables

**Figure 1 molecules-27-00051-f001:**
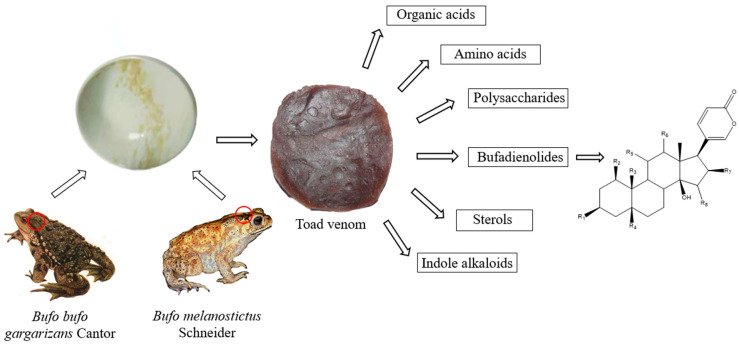
The preparation and chemical composition of toad venom.

**Figure 2 molecules-27-00051-f002:**
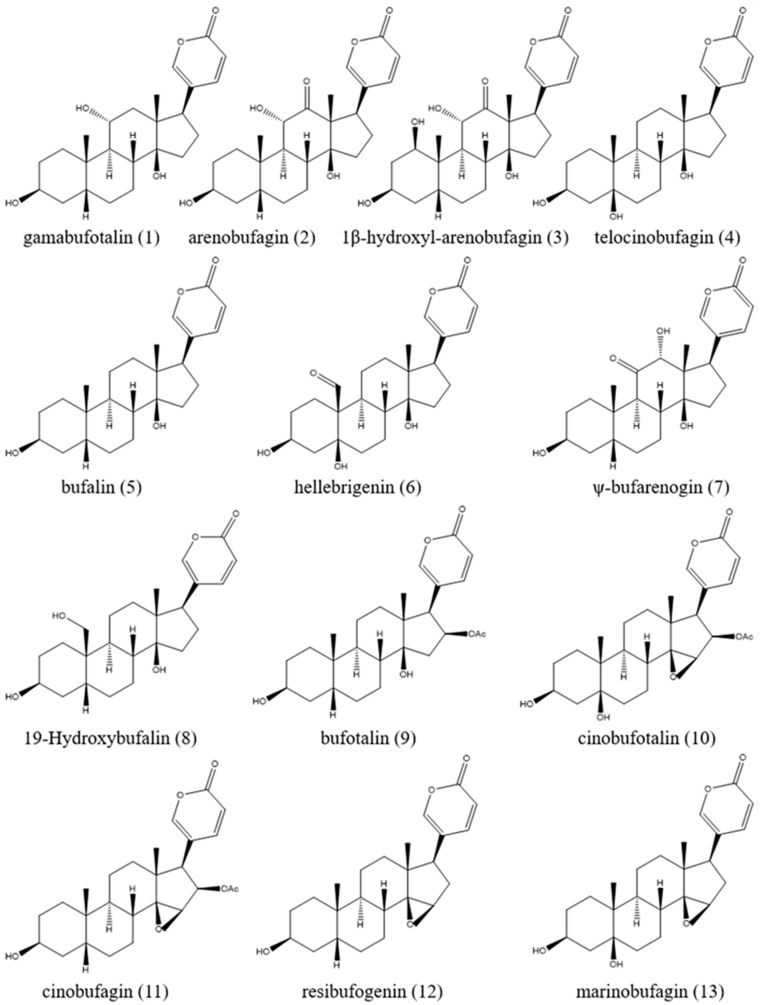
Chemical structure of the main bufadienolides.

**Figure 3 molecules-27-00051-f003:**
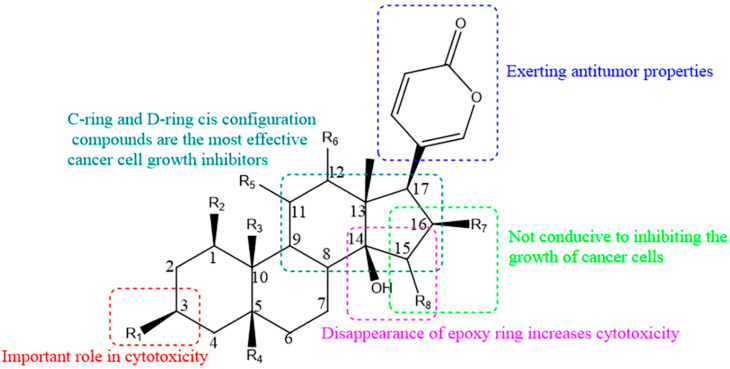
Structure–activity relationship diagram of bufadienolides.

**Figure 4 molecules-27-00051-f004:**
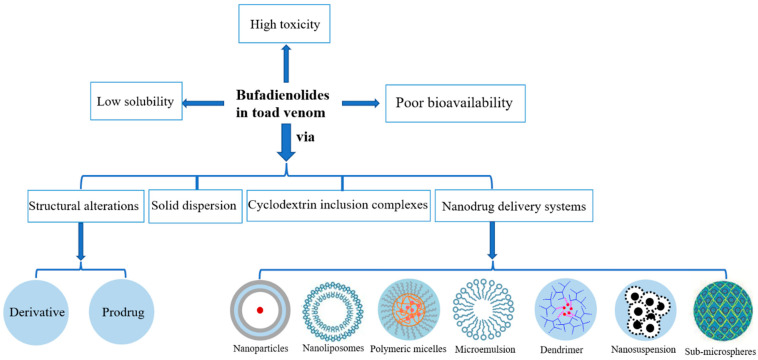
Strategies to improve solubility and bioavailability of bufadienolides.

**Figure 5 molecules-27-00051-f005:**
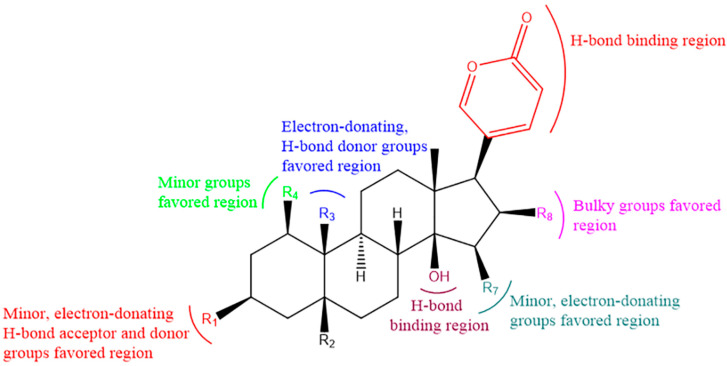
Structure–activity relationship of bufadienolides revealed by 3D-QSAR study.

**Figure 6 molecules-27-00051-f006:**
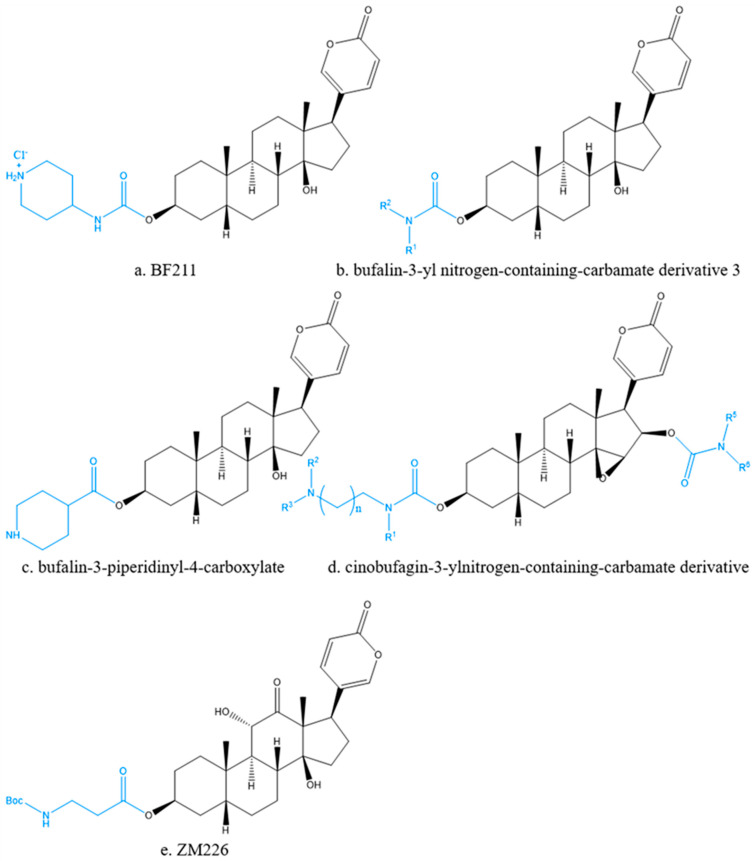
The derivatives of bufadienolides.

**Figure 7 molecules-27-00051-f007:**
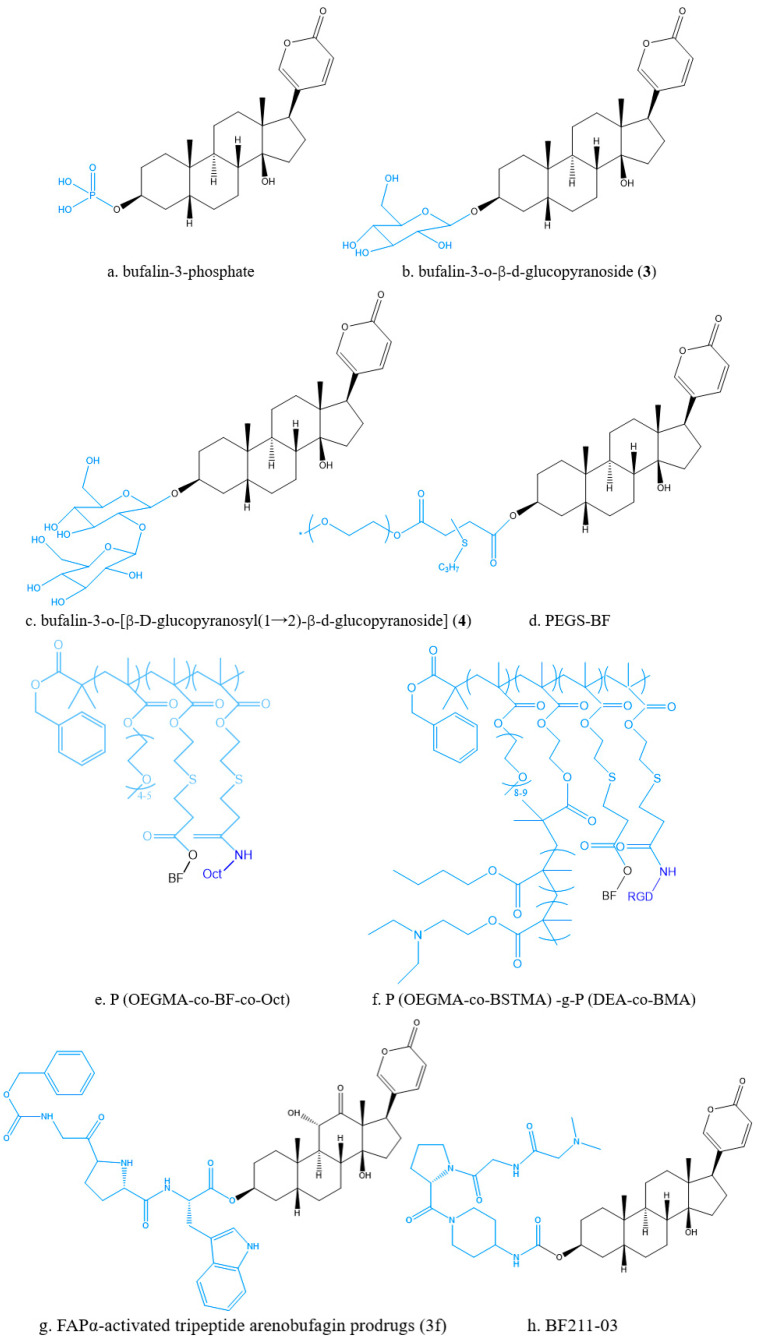
The prodrugs of bufadienolides.

**Figure 8 molecules-27-00051-f008:**
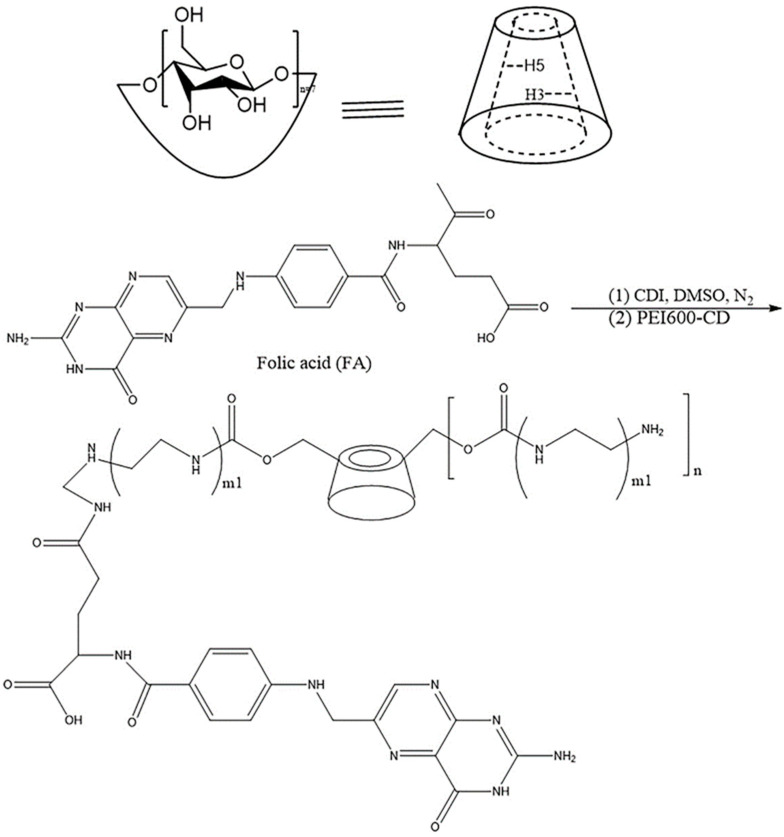
Chemical structures of β-cyclodextrin; the synthesis of FA-PEI-β-CD.

**Figure 9 molecules-27-00051-f009:**
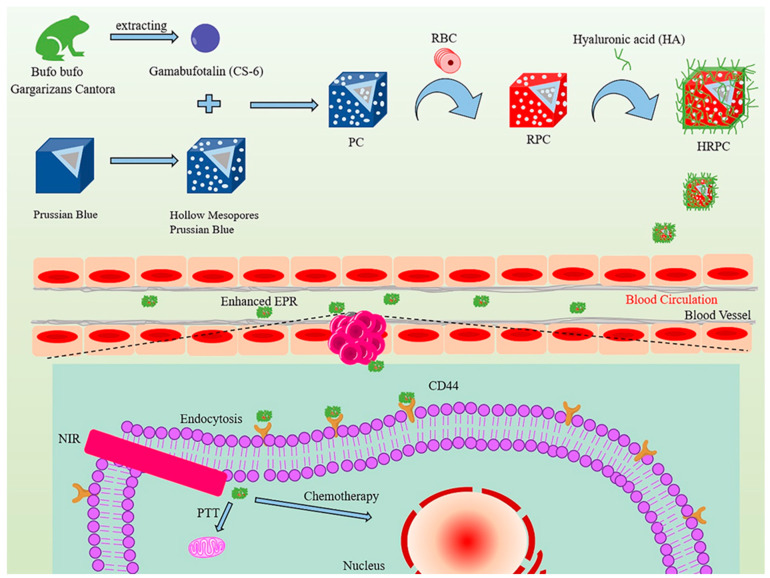
The illustration of a targeted nanodrug system loaded with CS-6 for breast cancer therapy, as well as its synthesis. Adapted from [116].

**Figure 10 molecules-27-00051-f010:**
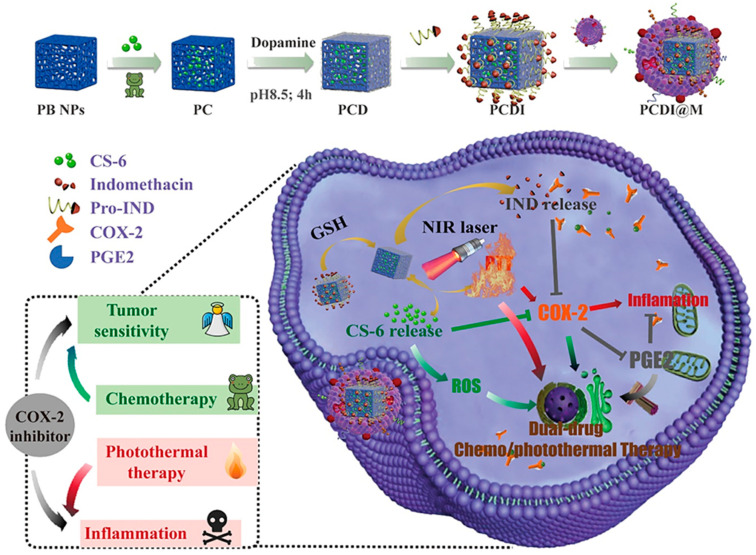
Synthesis process of PCDI@M nanocomplexes and the scheme of synergistic dual-drug chemo/photothermal therapy against cervical cancer. Adapted from [117].

**Figure 11 molecules-27-00051-f011:**
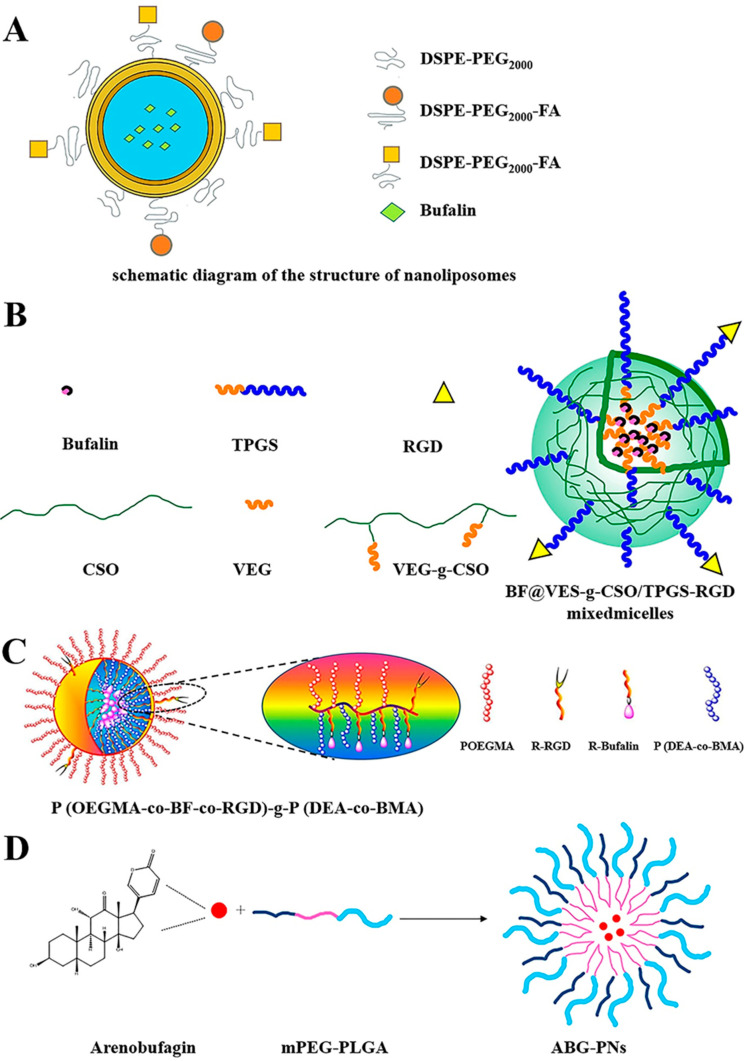
A schematic description of ABG-PNs’ preparation and structure (**A**). A schematic diagram of the structure of nanoliposomes (**B**). Formulation mechanism of Bu@Vec/t-rGd MM (**C**). Schematic for process of tumor-targeting micellar nanoparticles assembled from amphiphilic brush-type copolymers, P(OEGMA-co-BF-co-RGD)-g-P(DEA-co-BMA) (**D**). Adapted from [131].

**Figure 12 molecules-27-00051-f012:**
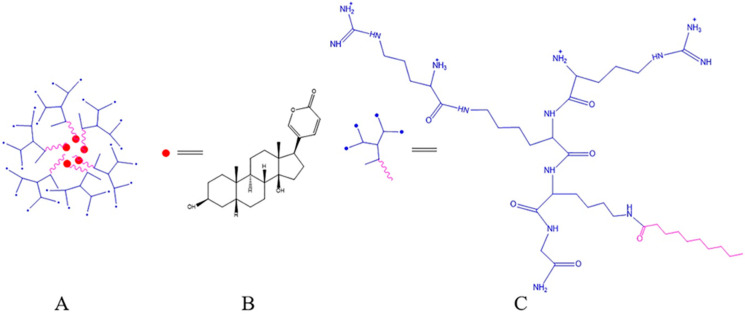
Molecular structure of bufalin-peptide-dendrimer inclusion (**A**); bufalin (**B**) and peptide-dendrimer (**C**).

## Data Availability

Not applicable.

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
