# Peer review of "Novel Strategies for Solubility and Bioavailability Enhancement of Bufadienolides"

_molecules, 2021, doi:10.3390/molecules27010051_

Round 1

Reviewer 1 Report

Very important review reporting the characteristics of bufadienolides and the pharmacological activities with a particular interest on anti-tumor effects. The authors also reported a very informative structure-activity relationship of the compounds which aims to better understand such molecules.

The review also described many strategies to improve solubility and bioavailability of bufadienolides in order to propose them in clinical settings and summarized structural modification, solid dispersion, cyclodextrin inclusion, microemulsion and nano-drug delivery system, including the use of nanoparticles. This part is very exhaustive and informative, showing the potential of these molecules as new anti-cancer drugs once exceed the limitation due to their low solubility.

Minor revision of English, for example, in Conclusion, line 6, …whether other ingredients are active, “ingredients” should be replaced by “compounds”.

Reference 27 has to be completed.

As the anti-tumor activity of bufadienolides is abundantly described in recent years, I encourage the authors to add the five following recent studies (2020-2021) reporting the diversity of anti-cancer effects of these compounds in additional cancer such as gastric, ovarian, pancreatic and colorectal cancers as well as melanoma.

Bufalin inhibits peritoneal dissemination of gastric cancer through endothelial nitric oxide synthase-mitogen-activated protein kinases signaling pathway. Zou D, Song J, Deng M, Ma Y, Yang C, Liu J, Wang S, Wen Z, Tang Y, Qu X, Zhang Y.FASEB J. 2021 May;35(5):e21601. doi: 10.1096/fj.202002780R.PMID: 33913201

Bufalin inhibits ovarian carcinoma via targeting mTOR/HIF-alpha pathway. Su S, Dou H, Wang Z, Zhang Q. Basic Clin Pharmacol Toxicol. 2021 Feb;128(2):224-233. doi: 10.1111/bcpt.13487. Epub 2020 Oct 2.

Toad Venom Antiproliferative Activities on Metastatic Melanoma: Bio-Guided Fractionation and Screening of the Compounds of Two Different Venoms. Soumoy L, Wells M, Najem A, Krayem M, Ghanem G, Hambye S, Saussez S, Blankert B, Journe F. Biology (Basel). 2020 Aug 10;9(8):218. doi: 10.3390/biology9080218.

Silencing c-Myc Enhances the Antitumor Activity of Bufalin by Suppressing the HIF-1alpha/SDF-1/CXCR4 Pathway in Pancreatic Cancer Cells. Liu X, Zhou Y, Peng J, Xie B, Shou Q, Wang J. Front Pharmacol. 2020 Apr 17;11:495. doi: 10.3389/fphar.2020.00495. eCollection 2020.

Bufalin reverses multidrug resistance by regulating stemness through the CD133/nuclear factor-kappaB/MDR1 pathway in colorectal cancer. Zhan Y, Qiu Y, Wang H, Wang Z, Xu J, Fan G, Xu J, Li W, Cao Y, Le VM, Ly HT, Yuan Z, Xu K, Yin P.Cancer Sci. 2020 May;111(5):1619-1630. doi: 10.1111/cas.14345. Epub 2020 Mar 16.

Reviewer 2 Report

The manuscript "Novel Strategies for Solubility and Bioavailability Enhancement of Bufadienolides" is essentially a review article. A significant part of the text is devoted to the description of the biological properties of bufadienolides. Therefore, according to the mind of reviewer, it would be advisable to expand its title to "Biological activity and strategies for solubility and bioavailability enhancement of bufadienolides".

For a more complete presentation of the material, it is required to bring the quantitative characteristics of the modified compositions - solubility, trans membrane permeability, bioavailability.  It seems to be represented at least in the form of Table contents.  Instead, a purely qualitative description is given without explaining the reasons for the change in the pharmacological parameters of the modified drugs.

In addition, given the review nature of the publication, it is advisable to supplement the text of the article with at least a brief description of the isolation and purification of noted biologically active substances from the body of animals. Also, a photo of this noted toad, as far as its Latin name, from which the described biologically active components are isolated, would not hurt as an illustration. Reviewer supposes, than noted information should be interesting for non-Chinese reader of your esteemed Journal.

According to the reviewer, the text of the article contains very little dosed of quantities description of biophysical results of modification of the pharmacological properties of Bufadienolides compositions. The Table 2 does not contain quantitative parameters of changes in solubility, trans-membrane permeability and bioavailability. However, this information, at least in a brief form, is necessary, according to the title of the manuscript and the declared goals of the research described in this review.
